# MADGEN: Mass-Spec attends to De Novo Molecular generation

**Yinkai Wang, Xiaohui Chen, Liping Liu, Soha Hassoun**[*]
Department of Computer Science
Tufts University
`{yinkai.wang, xiaohui.chen, liping.liu, soha.hassoun}@tufts.edu`

## Abstract

The annotation (assigning structural chemical identities) of MS/MS spectra remains a significant challenge due to the enormous molecular diversity in biological samples and the limited scope of reference databases. Currently, the vast majority of spectral measurements remain in the "dark chemical space" without structural annotations. To improve annotation, we propose MADGEN (Mass-spec Attends to De Novo Molecular GENeration), a scaffold-based method for de novo molecular structure generation guided by mass spectrometry data. MADGEN operates in two stages: scaffold retrieval and spectra-conditioned molecular generation starting with the scaffold. In the first stage, given an MS/MS spectrum, we formulate scaffold retrieval as a ranking problem and employ contrastive learning to align mass spectra with candidate molecular scaffolds. In the second stage, starting from the retrieved scaffold, we employ the MS/MS spectrum to guide an attention-based generative model to generate the final molecule. Our approach constrains the molecular generation search space, reducing its complexity and improving generation accuracy. We evaluate MADGEN on three datasets (NIST23, CANOPUS, and MassSpecGym) and evaluate MADGEN's performance with a predictive scaffold retriever and with an oracle retriever. We demonstrate the effectiveness of using attention to integrate spectral information throughout the generation process to achieve strong results with the oracle retriever. Our code is available at https://github.com/HassounLab/MADGEN

## 1 Introduction

Metabolomics, the measurement and identification of small molecules in biological samples, plays a critical role in numerous fields, including drug discovery, biomarker discovery, and environmental science. By analyzing the molecular composition of complex biological samples, metabolomics provides insights into cellular processes, metabolic pathways, and the effects of environmental changes on biological systems. Tandem mass spectrometry (MS/MS) has emerged as a powerful, widely used analytical technique that can separate and fragment molecules within a biological sample, thus producing rich spectra that can be further analyzed to annotate the measurements within the sample (Kind et al., 2018).

Despite the utility of metabolomics, assigning a chemical structural identity to a measured spectrum remains a significant challenge. Currently, most MS/MS spectra cannot be linked to known molecular structures due to the vast chemical diversity in biological samples and the limited scope of reference databases. Spectral databases that catalogue molecules and their measured spectra, e.g., MoNA (Davis) and GNPS (Wang et al., 2016), are used for identifying a close match to the measured spectra. However, such databases remain relatively small. Molecular databases such as PubChem (Kim et al., 2016) and KEGG (Kanehisa et al., 2021) are often utilized to provide *candidate* molecular structures when using computational methods such as SIRIUS (Dührkop et al., 2019), MLP or GNN-based approaches (Wei et al., 2019; Zhu et al., 2020) to predict the molecular structure that most likely produced the measured spectrum. Despite the success of these tools and the increased size of such databases, the "dark chemical space" of unknown molecules remains large,

---

[*]corresponding author

and hinders the interpretation of metabolomics data. De novo molecular structure generation from mass spectra is a promising approach to overcome the limitations of database-dependent methods. Further, de novo generation is crucial for discovering previously unknown molecules that play key roles in health, disease, and environmental processes.

Our insight in addressing this challenge herein is the use of scaffolds to simplify the structure generation process. A scaffold, or core structure, is used widely in medicinal chemistry to represent core structures of bioactive compounds (Hu et al., 2016). Such scaffolds can be modified with the addition of functional groups to enhance medicinal properties. By focusing on scaffold-based molecular generation in the context of annotation, we can reduce the complexity of structure generation and constrain the search space, making it more manageable and improving accuracy. Once a scaffold is predicted for a measured spectrum, it can guide the addition of structural elements (atoms and bonds) to the scaffold to generate the target molecule.

We propose a scaffold-based approach to de novo molecular structure generation guided by mass spectrometry data, with a focus on evaluating performance both when the scaffold is known and when it is predicted. Our contributions are as follows:

- We introduce a two-stage framework that first predicts a scaffold from the MS/MS spectrum, from which we then generate the target molecular structure. Given the challenges in accurately predicting the scaffold, we report performance under two settings: using the correct scaffold and using the predicted scaffold. This comparison highlights the potential and limitations of scaffold prediction in de novo molecular generation.

- Our method leverages fragmentation patterns in MS/MS spectra to guide scaffold prediction. While scaffold prediction is not always accurate, integrating even partially correct scaffolds reduces the complexity of de novo generation and constraints the search space to more plausible molecular structures.

- The scaffold-based design also improves interpretability, as even predicted scaffolds serve as structural anchors for understanding the generated molecules. This interpretability is crucial for analyzing potential biological functions and chemical properties in practical applications.

- Our approach has broad applicability in metabolomics, drug discovery, and environmental analysis, where the discovery of novel metabolites, bioactive molecules, and uncharacterized compounds is essential.

## 2 RELATED WORK

**De novo structure generation guided by mass spectra.** De novo molecular generation offers a promising alternative to database-dependent methods by directly (without the use of candidate molecules from databases) predicting or generating molecular structures from mass spectrometry data. MSNovelist (Stravs et al., 2022) relies on CSI:FingerID (Dührkop et al., 2015) to predict molecular fingerprints from the query mass spectrum, and then uses a LSTM model to reconstruct molecules. Spec2Mol (Litsa et al., 2023) employs a convolutional neural network to map MS/MS spectra to a latent space, generating molecular structures as SMILES strings. MassGenie (Shrivastava et al., 2021) uses a transformer-based model trained on real and synthetic spectra to generalize to unseen compounds, leveraging transformers' strength in handling sequential data. MS2Mol (Butler et al., 2023) extends these approaches with a transformer-based encoder-decoder, incorporating byte-pair encoding and precursor mass, to improve accuracy. There were no consistent datasets that were used to evaluate these models. For example, MSNovelist is evaluated on 3,863 MS/MS spectra from the GNPS library (Wang et al., 2016), while Spec2Mol is evaluated on the NIST2020 dataset. Further, not all these tools are available in the public domain. Recently, The MassSpecGym dataset (Bushuiev et al., 2024) was developed as a benchmark dataset to standardize the evaluation on de novo generation, retrieval, and spectra simulation tasks. We utilize this dataset, and two others, to report the performance of MADGEN. We also compare our results with the best reported results so far on the MassSpecGym dataset.

**Generative frameworks for molecular generation.** Generative models have become essential in molecular generation due to their ability to approximate complex distributions in the chemical space. These models, such as VAEs, GANs, and Diffusion models, treat molecules as graphs, enabling

them to capture the relational properties between atoms and bonds (Zhu et al., 2022). Structure-constrained molecular design is a key strategy in modifying an existing candidate structure with the goal of attaining improved molecular properties. A common approach is constraining molecular generation to contain a specific scaffold or a molecular fragment, e.g., Podda et al. (2020), Li et al. (2019), Green et al. (2021). These models often allow for an arbitrary scaffold as an initial structure that captures a desired property. Unlike these models, MADGEN employs Murcko scaffolds (Bemis & Murcko, 1996), a standard scaffold used across many chemical and biological studies due to its ability to represent the core backbone of molecules. As there are currently no methods to predict this scaffold for a query spectra, the first step of MADGEN predicts the scaffold from a list of candidate molecules. Importantly, generative models have shown value in exploring the uncharacterized chemical spaces (Holdijk et al., 2022; Chen et al., 2023; Duan et al., 2024). For example, Retro-Bridge (Igashov et al., 2023) models the dependencies between the spaces of substrate and product molecules in the context of chemical reactions as a stochastic process between two distributions. RetorBridge uses a Markov bridge process to approximate dependencies between these intractable distributions. RetroBridge is adapted for MADGEN's second step, where we aim to model the joint distribution of scaffolds and target molecules, and starting with the Murcko scaffold, we utilize the mass spectrum to guide the generation process towards the target molecule.

## 3 METHODS

Direct generation of molecules from mass spectra is a hard problem. In this work, we propose to divide the problem into two simpler sub-problems (see Figure 1): we first retrieve the molecular scaffold from the mass spectrum and then generate the target molecule conditioned on both the mass spectrum and the scaffold. We conjecture that the scaffold prediction problem is easier than predicting the target molecule because the scaffold usually has a simpler structure than the target molecule. Consequently, the molecule generation task becomes easier when the scaffold is known.

### 3.1 SCAFFOLD RETRIEVAL

The goal of scaffold retrieval is to identify the scaffold of the target molecule. Denote an MS/MS spectrum and its chemical formulate as $X = (X^{\mathrm{ms}}, X^{\mathrm{cf}})$. Scaffold retrieval takes $X$ as input and retrieves the core scaffold that represents the fundamental backbone of the molecule, including its ring systems and central framework. With a correct scaffold served as the starting point for further molecular generative process, the complexity of the search space is significantly reduced.

However, predicting the scaffold from spectral data is a challenging problem due to the non-linear relationship between fragmentation patterns and the scaffold substructures. In this work, we explore two scaffold retrieval strategies - predictive retrieval and oracle retrieval.

**Predictive retrieval.** We formulate the scaffold retrieval as a ranking problem. Given a set of scaffold candidates $\mathbb{S}$, the goal is to use a neural network to score each candidate $S \in \mathbb{S}$ given $X$ such that scaffold with highest score $S^*$ maximally resembles the correct scaffold $S^{\mathrm{gt}}$. We rank on a candidate set from which the target molecules have been removed, introducing the possibility that the true scaffold may not be present in the set.

A straightforward approach is to directly train a binary classifier that tells whether the given pair $(X, S)$ is matched or not. However, to fully leverage the relationship between the spectrum and scaffold modalities, we adopt a contrastive learning framework similar to CLIP (Radford et al., 2021). In this framework, the spectrum $X$ is treated as one modality, while the scaffold $S$ is treated as the other. Contrastive learning aligns the embeddings of these two modalities in a shared latent space, enabling the model to learn a meaningful representation of their relationships.

This paradigm has been widely employed in multimodal information retrieval frameworks (Luo et al., 2021; Bain et al., 2022; Lei et al., 2021; Fang et al., 2021; Ma et al., 2022; Hendriksen et al., 2022), where embedding similarity is used to determine the most likely paired item based on a query. Similarly, in our framework, we align the embeddings of mass spectra and scaffolds to facilitate scaffold retrieval. Specifically, we employ contrastive learning techniques inspired by JESTR (Kalia et al., 2024), which was designed to align the embeddings of mass spectra with their corresponding molecules.

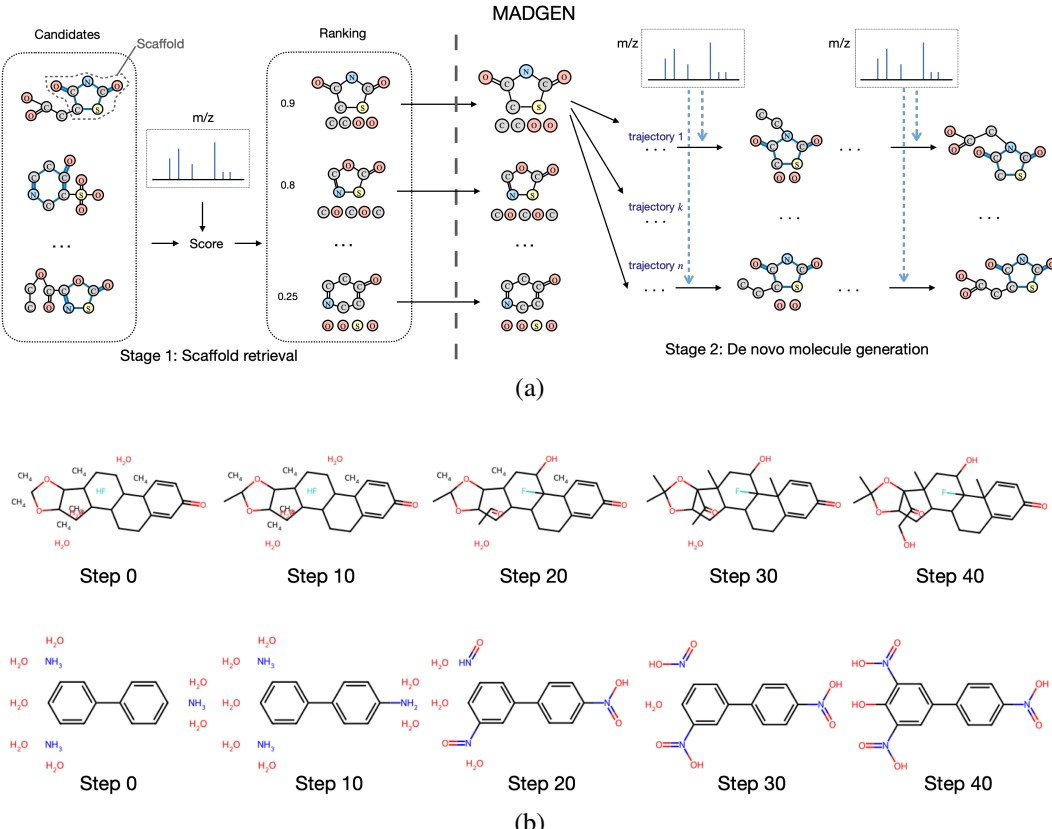

Figure 1: MADGEN overview and example. (a) The overview of MADGEN. The mass spectra are used to rank scaffold candidates through contrastive learning. The top-ranked scaffold, with blue edges fixed, serves as a foundation for de novo molecule generation, guided by the spectra at each generation step. (b) Examples of molecular generation process over time steps for Kenalog from CANOPUS dataset (upper) and 2,6-Dinitro-4-(4-nitrophenyl)phenol from NIST23 dataset (lower). The scaffolds remain fixed, while additional edges are introduced in each step to connect free atoms to scaffolds. The complete molecules are shown in step 40.

To achieve this alignment, we introduce two separate encoders to project the mass spectra and scaffold graphs into a shared latent space. Specifically, the mass spectra $X$ are projected using a multi-layer perceptron (MLP) encoder $f_X$, which maps the spectral data into a $d$-dimensional latent space. We employ a graph neural network (GNN) $f_S$ to encode the global representation of the scaffold graph $S$, also into a $d$-dimensional space.

The key insight of this approach is to ensure that the embeddings of the matched spectrum and scaffold are close to each other in the latent space. Both encoders, $f_X$ and $f_S$, are jointly trained using a contrastive learning objective. This objective ensures that the embeddings of matched spectrum-scaffold pairs are close in the joint embedding space, while mismatched pairs are pushed apart. Specifically, for each spectrum $X$ and scaffold $S$, we compute a similarity score, $h(z_{\text{spec}}^n, z_{\text{mol}}^m)$, defined as:

$$h(z_{\text{spec}}^n, z_{\text{mol}}^m) = \exp\left(\frac{z_{\text{spec}}^n \cdot z_{\text{mol}}^m}{\|z_{\text{spec}}^n\|\|z_{\text{mol}}^m\|\tau}\right), \tag{1}$$

where $z_{\text{spec}}^n$ and $z_{\text{mol}}^m$ are the embeddings of the spectrum and molecular scaffold, respectively, and $\tau$ is a temperature hyperparameter that controls the importance of non-matching pairs.

The contrastive loss $\mathcal{L}_{\text{contrastive}}$ is computed over a batch of size $k$ as:

$$\mathcal{L}_{\text{contrastive}} = \frac{1}{k} \sum_{n=1}^{k} \left[ -\mathbb{E} \left[ \log \frac{h(z_{\text{spec}}^n, z_{\text{mol}}^n)}{\sum_{m=1}^{k} h(z_{\text{spec}}^n, z_{\text{mol}}^m)} \right] \right]. \tag{2}$$

Here the two embeddings in the numerator are from the same molecule $n$ while the two in the denominator are from two different molecules. This loss encourages the model to assign high values to matching spectrum-scaffold pairs and lower values to non-matching pairs, effectively aligning the embeddings in the latent space.

After training, we access the score of each scaffold candidate via cosine similarity between the mass spectra embedding $f_X(X)$ and the scaffold embedding $f_S(S)$. To improve scaffold ranking accuracy, we introduce a frequency-based aggregation approach. For each data point, we first retrieve the top-$k$ ranked candidate scaffolds. The frequency of each scaffold appearing in these top candidates is then computed, and the most frequently occurring scaffold for each formula is selected as the predicted scaffold. This method refines scaffold selection by leveraging consensus among top-ranked candidates, leading to improved scaffold prediction accuracy (SPA).

**Oracle retrieval.** We maintain a lookup table as an oracle which always yields the correct scaffold given the MS/MS spectrum and the chemical formula. We construct the lookup table by extracting the scaffold from the molecular graph representation using RDKit. This lookup table serves as an idealized oracle, simulating perfect scaffold retrieval. It allows us to focus on assessing the second stage of molecular generation: the task of adding side chains and functional groups to the scaffold-independently from any potential errors that could occur in scaffold retrieval.

## 3.2 SCAFFOLD-CONDITIONED DE NOVO MOLECULE GENERATION WITH SPECTRA GUIDANCE

### 3.2.1 NOTATIONS AND PROBLEM FORMULATION

We represent a molecule $G$ as a graph $G = (\mathcal{V}, \mathcal{E})$. Its scaffold $S = (\mathcal{V}^S, \mathcal{E}^S)$ is a subgraph of $G$. Since the atom set $\mathcal{V}$ can be directly inferred from the chemical formula, the task of molecular generation involves determining the appropriate edge set $\mathcal{E} \setminus \mathcal{E}^S$ that connects the scaffold to the remaining isolated atoms $\mathcal{V} \setminus \mathcal{V}^S$. While there are combinatorially many valid edge sets that could complete the molecule from the scaffold, we utilize spectral data $X$ to guide the edge generation process and ensure the structure aligns with the observed spectra.

### 3.2.2 SCAFFOLD-CONDITIONED GENERATION VIA MARKOV BRIDGE

We frame the molecule prediction task as generating graphs given a scaffold. Specifically, starting from a scaffold $S$, we are interested in modeling the distribution $p(G|S) = p(\mathcal{E}|\mathcal{E}^S, \mathcal{V}^G)$ with the following Markov decomposition:

$$p(\mathcal{E}|\mathcal{E}^S, \mathcal{V}^G) = \sum_{\mathcal{E}_0:\mathcal{E}_{T-1}} \prod_{t=0}^{T-1} p(\mathcal{E}_{t+1}|\mathcal{E}_t, \mathcal{E}^S, \mathcal{V}^G), \tag{3}$$

where $\mathcal{E}_0 = \emptyset$ can be considered the case where no bonds are formed from isolated atoms to others, and $\mathcal{E}_T = \mathcal{E}$. The sequence of random variables $\mathcal{E}_{0:T}$ can be viewed as progressively connecting atoms to form the final molecules.

Let $e_t$ be an arbitrary edge entry in $\mathcal{E}_t$, $e_t$ can be represented as a D-dimensional one-hot vector, with 0 class being non-edge and 1 to D-1 classes being the bond types. Following Austin et al. (2021), we formulate the transition probabilities $p(e_{t+1}|e_t, e_T)$ conditioned on the endpoint $e_T$:

$$p(e_{t+1}|e_t, e_T) = \text{Categorical}(e_{t+1}; \mathbf{Q}_t(e_T)e_t), \tag{4}$$

where $\mathbf{Q}_t(e_T) \in \mathbb{R}^{D \times D}$ is an absorbing transition matrix conditioned on the endpoint data $e_T$ (Igashov et al., 2023).

With the defined model, we now approximate it with a parameterized distribution:

$$p_\theta(e_{t+1}|e_t, \mathcal{E}^S, \mathcal{V}^G) = \text{Categorical}(e_{t+1}; \mathbf{Q}_t(\hat{e}_T)e_t), \text{ where } \hat{e}_T = \text{nn}_\theta(\mathcal{E}_t, \mathcal{E}_S, \mathcal{V}^G) \tag{5}$$

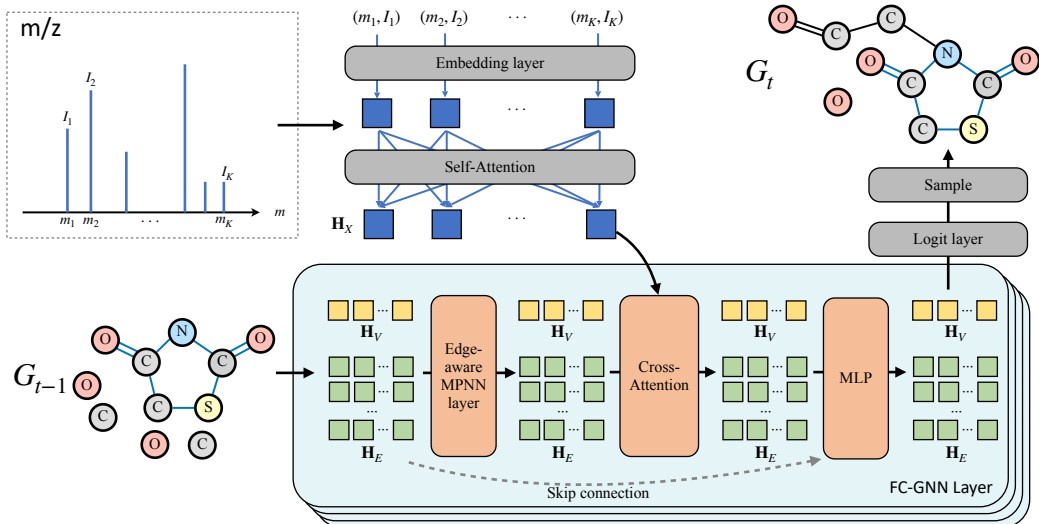

Figure 2: Overview of the MADGEN model framework. The input consists of m/z peaks and intensities $(m, I)$, which are passed through an MLP for embedding. These embeddings are processed through self-attention and combined with the molecular graph's node and edge embeddings via cross-attention. The node and edge embeddings are updated iteratively using an edge-aware message-passing neural network(MPNN) and fully-connected graph neural network (FC-GNN) layers. The final molecular structure is sampled after the last time step via a logit layer, aligning with the mass spectral data.

is the endpoint prediction via a neural network $\text{nn}_\theta(\cdot)$. Given a pair $(S, G)$ from the dataset, we train $\text{nn}_\theta(\cdot)$ by optimizing the evidence lower bound (ELBO)

$$\mathcal{L}_\theta(S, G) := -T\mathbb{E}_{\mathcal{U}(t;0,T-1)}\mathbb{E}_{p(e_t|e_0,e_T)}\Big[\text{KL}\Big(p\big(e_{t+1}\big|e_t, e_T\big)\big\|p_\theta\big(e_{t+1}\big|e_t, \mathcal{E}^S, \mathcal{V}^G\big)\Big)\Big]. \quad (6)$$

Here, $p(e_t|e_0, e_T)$ represents the probability of transitioning to an arbitrary timestep $t$ from $T$, which can be expressed in a closed form. The detailed derivation of the ELBO and the transition distributions is provided in Appendix A.3.

To obtain pairs $(S, G)$ for training, we first randomly sample a graph $G$ from the data distribution $p(G)$. The scaffold $S$ of $G$ is computed using RDKit. This results in the joint data distribution $p(S, G) = p(G)p(S|G)$, where $p(S|G)$ is a Dirac delta distribution that assigns all its probability mass to the scaffold $S$ derived from $G$.

### 3.2.3 CLASSIFIER-FREE GUIDANCE FROM MASS SPECTRUM

We introduce the mass spectrum $X^{\text{ms}}$ as an additional conditioning term to refine the search space during the generation of $G$ from $S$. The neural network $\text{nn}_\theta(\cdot)$ is designed to condition on $X$ when computing the logits.

To integrate spectrum information throughout the generation process, we utilize classifier-free guidance (CFG) (Ho & Salimans, 2022). At each inference step, for each edge entry, we compute the logit $\ell_c$ conditioned on the spectrum $X$, and the logit $\ell_u$ without conditioning. The final logit $\ell_g$ is then obtained by combining the two using a guidance scale $\lambda_t$:

$$\ell_g = (1 + \lambda_t)\ell_c - \lambda_t\ell_u. \quad (7)$$

During training, we randomly remove the spectrum condition with a probability of 0.1 to enable CFG. Since CFG tends to prioritize generation quality over diversity, increasing $\lambda_t$ helps reduce the search space and improves the success rate of generating target molecules based on the given spectrum.

We provide further details on how the CFG techniques are integrated into our framework (see Figure 2), particularly within the network architecture $\text{nn}_\theta(\cdot)$. We treat the graph as fully connected,

| Dataset | #Spec. | #Mol. | #Scaf. | #Free atoms | per Mol. | | | per Scaf. | |
|---------|--------|-------|--------|-------------|----------|----------|----------|-----------|--------|
| | | | | | #Node | #Edge | #Spec. | #Node | #Mol. |
| NIST23 | 689,358 | 40,934 | 13,560 | 6.6 | 21.0 | 22.4 | 16.84 | 14.4 | 3.02 |
| CANOPUS | 6,618 | 6,618 | 3,137 | 8.5 | 28.6 | 31.0 | 1.00 | 20.1 | 2.11 |
| MassSpecGym | 231,104 | 31,602 | 15,649 | 10.9 | 25.8 | 27.4 | 6.60 | 22.1 | 2.02 |

Table 1: Statistics on the three datasets, including the number of molecules (Mol.), spectra (Spec.), scaffolds (Scaf.), average number of free atoms, and average statistics per molecule and per scaffold.

where non-edges are considered a specific type of edge, and apply a fully connected graph neural network (FC-GNN) to compute on this structure. The detailed design of the FC-GNN is provided in the Appendix A.1. Two key components to highlight are the encoding of the mass spectrum $X$ and the conditioning mechanism.

**Mass spectrum encoding as tokenization.** We represent the $X$ as a set of peaks $\{P_1, \ldots, P_K\}$, where $P_k = (M_k, I_k)$ is the m/z and intensity values. We encode each peak into a embedding vector via an MLP. We then use a self-attention module to boost the information flow among the peak representations. The full computation is as follow:

$$\mathbf{H}_X = \text{Self-Attention}(\mathbf{h}'_1, \ldots, \mathbf{h}'_K), \ \mathbf{h}'_k = \text{concat}(\text{MLP}(M_k), \ \text{MLP}(I_k)). \tag{8}$$

This approach results in a variable-length representation of the mass spectrum, $\mathbf{H}_X$, where each peak representation $\mathbf{h}_k$ aligns with potential substructures in the molecule. By retaining these individual peak representations, the model is better able to guide the generation of subgraphs that correspond to molecular fragments consistent with the observed spectral data.

**Spectrum conditioning via cross-attention.** We map $\mathbf{H}_X$ to each message passing layer of FC-GNN via cross-attention. Since there are intermediate representations for both nodes and edges, we explore three cross-attention paradigms: node-only attention, edge-only attention and both. We replace the $\mathbf{H}_X$ with a learnable embedding when the spectrum data are removed.

## 4 EXPERIMENTS

### 4.1 DATASETS

We evaluate the performance of MADGEN on three datasets (Table 1). The NIST23 dataset (National Institute of Standards and Technology (NIST), 2023) is curated by the National Institute of Standards and Technology to provide reference spectral data for a wide range of chemical molecular standards to support research and development. It is available for purchase. Each molecule is measured using various mass spectrometry instruments, and under various instrument settings, thus contributing to the high number of spectra/molecule. The CANOPUS dataset is the smallest dataset, and it was designed to train and evaluate the CANOPUS tool (Dührkop et al., 2021), which predicts compound classes, e.g., alcohols, phenol ethers, and others, from spectra. It has a 1:1 spectra to molecule ratio. It was used recently to benchmark other metabolomics tools such as MIST(Goldman et al., 2023) and ESP(Li et al., 2024). The newly developed MassSpecGym benchmark dataset (Bushuiev et al., 2024) is collected from many public reference spectral databases and curated uniformly. The MassSpecGym is the largest publicly available labeled mass spectra dataset. For all three datasets, few molecules shared the same scaffold.

All datasets were preprocessed by normalizing the intensities of the MS/MS spectra and removing low-intensity peaks below a predefined threshold to reduce noise. The NIST23 and CANOPUS datasets were split into training, validation, and test sets based on the scaffold, ensuring that scaffolds are unique to each split. This split prevents data leakage and ensures robust evaluation of model performance. For MassSpecGym, we utilized the split suggested by the benchmark (Bushuiev et al., 2024), which is based on the Maximum Common Edge Substructure (MCES). This split allows assessing the model generalization on novel molecules.

## 4.2 Experimental Setup and Evaluation Metrics

The model was trained using a graph transformer with 5 layers and 50 diffusion steps. We employed the AdamW optimizer with a learning rate of $1 \times 10^{-5}$. Full training details and hyperparameters can be found in Appendix A.2. For candidate pool selection, the following approaches were employed:

- **NIST23 and CANOPUS**: all candidate molecules were retrieved from PubChem using the chemical formula as a query, ensuring comprehensive coverage of possible structures.

- **MassSpecGym**: the candidate pool consists of 256 molecules for each test molecule. These candidates are selected based on the molecular formula provided by the MassSpecGym dataset. The target molecule is removed from the candidate pool.

The performance of the model is evaluated using the following metrics endorsed for model evaluation for the MassSpecGym benchmark:

- **Top-$k$ accuracy:** Measures the likelihood of the generating the true target structure among the top-$k$ generated molecules. We report the results for $k=1,10$. The generated molecules are ranked by the probabilistic nature of the model.

- **Tanimoto Similarity:** This metric evaluates the similarity between the generated structures and the ground truth molecules using molecular fingerprints. Higher Tanimoto similarity indicates that the predicted structure closely resembles the correct structure. We extracted fingerprint representations based on the Morgan algorithm (Morgan, 1965) using the RDKit toolkit (RDKit, online). The Morgan fingerprints are computed for radius 2 and 2048 bits.

- **Maximum Common Edge Substructure (MCES):** This metric is the edit distance between two molecules, and reflects the similarity of the largest common substructure between generated and ground truth molecules(Kretschmer et al., 2023).

- **Scaffold Prediction Accuracy (SPA):** In the scaffold prediction task, we assess how well the model predicts the core scaffold of the molecule compared to the ground truth scaffold.

## 4.3 Results

Our experiments, summarized in Table 2 , evaluate model performance on three datasets: NIST23, CANOPUS, and MassSpecGym, using both predictive and oracle retrievers. For the scaffold prediction task, we report a Scaffold Prediction Accuracy (SPA) for the predictive retriever ranging from 13.2% to 40.3%. Notably, the NIST23 dataset achieves the highest SPA of 40.3%, reflecting its lower scaffold diversity compared to CANOPUS and MassSpecGym, which have more complex scaffolds.

The metrics for the scaffold-based generation task reveal that the low scaffold prediction accuracy of the predictive retriever constrains molecular generation performance. For instance, on the NIST23 dataset, the predictive retriever yields a top-1 accuracy of 4.6%, while for CANOPUS and MassSpecGym, the top-1 accuracies is 2.10 % and 1.31%. Despite these challenges, the predictive retriever demonstrates moderate performance improvements compared to baseline methods like Spec2Mol and random generation.

In contrast, the oracle retriever, which has access to the correct scaffold, dramatically boosts performance. On NIST23, MADGEN achieves a top-1 accuracy of 49.0% and a top-10 accuracy of 65.5%, demonstrating the model's capacity to generate accurate molecular structures if the scaffold is known. Similarly, when using the oracle retriever, the performance on CANOPUS and MassSpec-Gym is significantly higher than the predictive retriever, with top-1 accuracies of 18.7% and 10.5%, respectively, showing the clear advantage of having access to correct scaffold information. Importantly, MADGEN outperforms the best published state-of-the-art performance (last row in Table 2) reported for the MassSpecGym dataset (Bushuiev et al., 2024) when using random chemical generation. The high top-1 and Top10 accuracy for the NIST23 dataset can be attributed to its smaller number of free atoms. MADGEN's task of completing the target molecule by adding edges to these free atoms is easier with a smaller number of free atoms. CANOPUS has the highest number of average free atoms, and the lowest top-1 and top-10 accuracies.

Baseline methods like Spec2Mol and MSNovelist are also included in the comparison. As shown in Table 2, MSNovelist results are limited to accuracy metrics, as other measures are not available. The

"-" in the table denotes this lack of data, while underlined values highlight the best results achieved by predictive retrievers, serving as a benchmark against the oracle retriever.

| Retriever | SPA↑ | Top1 | | | Top10 | | |
|---|---|---|---|---|---|---|---|
| | | Accuracy↑ | Similarity↑ | MCES↓ | Accuracy↑ | Similarity↑ | MCES↓ |
| NIST | | | | | | | |
| Spec2Mol | - | 0.0% | 0.10 | 20.88 | 0.0% | 0.12 | 13.66 |
| MSNovelist | - | 0.0% | - | - | 0.0% | - | - |
| MADGENPred. | 40.3% | 4.6% | 0.11 | 72.38 | 7.3% | 0.18 | 69.34 |
| MADGENOracle | 100% | **49.0%** | **0.63** | **18.48** | **65.5%** | **0.80** | **3.88** |
| CANOPUS | | | | | | | |
| Spec2Mol | - | 0.0% | 0.09 | 38.97 | 0.0% | 0.14 | 23.97 |
| MSNovelist | - | 0.0% | - | - | 0.0% | - | - |
| MADGENPred. | 17.7% | 2.10% | 0.22 | 20.56 | 2.39% | 0.27 | 12.69 |
| MADGENOracle | 100% | **18.7%** | **0.51** | **9.72** | **22.2%** | **0.59** | **4.44** |
| MassSpecGym | | | | | | | |
| Rand. Gen. | - | 0.0% | 0.08 | 21.11 | 0.0% | 0.11 | 18.25 |
| SMILES Transformer | - | 0.0% | 0.03 | 79.39 | 0.0% | 0.10 | 52.13 |
| SELFIES Transformer | - | 0.0% | 0.08 | 38.88 | 0.0% | 0.13 | 26.87 |
| Spec2Mol | - | 0.0% | 0.09 | 45.89 | 0.0% | 0.13 | 32.60 |
| MSNovelist | - | 0.0% | - | - | 0.0% | - | - |
| MADGENPred. | 13.2% | 1.31% | 0.20 | 27.47 | 1.54% | 0.26 | 16.84 |
| MADGENOracle | 100% | **10.5%** | **0.43** | **16.27** | **12.4%** | **0.53** | **7.08** |

Table 2: Performance metrics for various datasets using both predictive and oracle retrievers. The table presents top-1 and top-10 accuracy, Tanimoto similarity, and Maximum Common Edge Substructure (MCES) scores. Best performance for each dataset is **bold**. The second-best performance for each dataset is underlined

## 4.4 ABLATION STUDY ON CONDITIONING MECHANISM

We conducted an ablation study to assess the impact of different encoding strategies, conditioning methods, and the use of CFG (Table 3) on the performance of MADGEN. Conditioning using the tokenization + cross-attention mechanism significantly improves the model performance. We believe this is because such mass spectra encoding is more efficient in encoding peak information without further compression. Further, through cross-attention, nodes and edges are able to query peaks of relevant importance. Importantly, upon introducing the self-attention into the mass spectrum encoder, a dramatic performance gain is observed. The self-attention significantly enhances mass spectra representations. We observed further performance gains using CFG on node or node and edge (both). Node-only CFG yields the best performance among all settings.

| Encoding strategy | Conditioning strategy | Accuracy | Similarity | MCES |
|---|---|---|---|---|
| Binning + MLP | Concatenation | 4.30% | 0.068 | 87.89 |
| Tokenization | Cross-Attn | 13.0% | 0.304 | 55.25 |
| Tokenization + Self-Attn | Cross-Attn | 42.5% | 0.642 | 23.90 |
| Tokenization + Self-Attn | Cross-Attn + CFG (edge) | 42.0% | 0.632 | 25.01 |
| Tokenization + Self-Attn | Cross-Attn + CFG (node) | **49.0%** | **0.694** | **18.48** |
| Tokenization + Self-Attn | Cross-Attn + CFG (both) | 45.9% | 0.667 | 21.89 |

Table 3: Ablation study results comparing different encoding strategies (Binning + MLP, Tokenization, Tokenization + Self-Attention) and conditioning strategies (Concatenation, Cross-Attention, Cross-Attention + CFG). The metrics evaluated are Accuracy (%), Tanimoto Similarity, and Maximum Common Edge Substructure (MCES). The best results were obtained using Tokenization + Self-Attention with Cross-Attention + CFG (node).

## 4.5 SENSITIVITY ANALYSIS OF FREE ATOM NUMBERS ON ACCURACY

We analyze how the number of free atoms will affects the generation accuracy of MADGEN. Figure 3 shows MADGEN's accuracy@1 and accuracy@10 on different number of free atoms across three datasets. We can observe that having more free atoms yields worse predictive accuracy, which is as expected as the learning complexity increases.

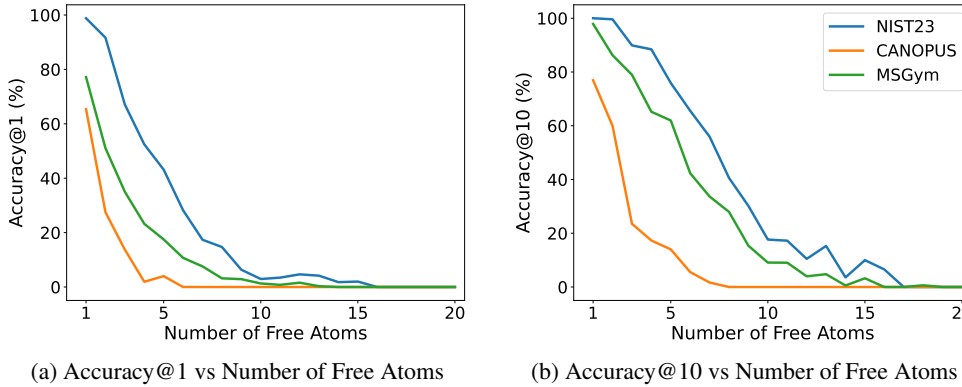

(a) Accuracy@1 vs Number of Free Atoms          (b) Accuracy@10 vs Number of Free Atoms

Figure 3: Accuracy vs Number of Free Atoms: With more free atoms for MADGEN to connect to the scaffold, the complexity of the generative trajectory increases, leading to a worse predictive accuracy.

## 5 CONCLUSION & FUTURE WORK

De novo annotation of mass spectrometry data is notoriously difficult, with a current best accuracy of 0% on the MassSpecGym dataset. MADGEN offers a novel two-stage framework for spectra-guided de novo annotation. The first stage, scaffold retrieval, is a new problem formulation whose solution can provide partial insight in regard to the molecular backbone of the measured spectra. Such insights may shed light on the molecule's class or properties. Our results show that this problem is challenging, achieving a scaffold prediction accuracy of 13.2%-40.3% for the three datasets. The second stage, de novo generation from an existing scaffold showed excellent results when using an oracle scaffold predictor, achieving an accuracy of 10.5%-49% across the three dataset. For the MassSpecGym benchmark, we achieved an accuracy of 2.10% and 1.31%. As with other tools, e.g., (Goldman et al., 2023), we conjecture that performance of MADGEN can be increased by incorporating additional data in the form of peak chemical formulae or molecular properties that correlate with fragmentation patterns. Potentially, the scaffold problem can be made easier if larger more distinct scaffold structures were utilized instead of the Murcko scaffold used herein. A bigger scaffold can in turn facilitate the de novo generation task. Further, an end-to-end MADGEN may reduce the compounding of errors across the two stages.

### ACKNOWLEDGMENTS

Research reported in this publication was supported by the National Institute of General Medical Sciences of the National Institutes of Health under award number R35GM148219. The content is solely the responsibility of the authors and does not necessarily represent the official views of the NIH. Chen and Liu are supported by the NSF CAREER Award 2239869.

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

# A    Appendix

## A.1    Model Architectures and Algorithms

In this section, we describe the architecture of our proposed scaffold-conditioned molecular generation guided by mass spectra data. The process consists of two main stages: scaffold retrieval (Stage 1) and scaffold-conditioned molecular generation (Stage 2). The model integrates node, edge, and spectral features, updated iteratively through cross-attention and self-attention mechanisms.

### A.1.1    Stage 1: Scaffold Retrieval

The first stage of the process involves predicting a scaffold that is most consistent with the input MS/MS spectrum. This stage utilizes a contrastive learning framework that aligns molecular graphs with their corresponding mass spectra. We use the 'MLP_BIN' model for encoding the spectral data.

**Molecular encoder.**    We employ a Graph Neural Network (GNN) to encode the molecular structures:

- **Node Features:** The molecular graph nodes (atoms) are encoded using GNN layers, where each node is associated with a feature vector that encodes atom type and other properties.

- **Edge Features:** Bonds between atoms are represented by edge features, which are also encoded by the GNN.

- **Graph Pooling:** The output node embeddings from the GNN are pooled using a MaxPooling layer to create a graph-level representation.

**Spectral encoder (MLP_BIN).**    For encoding the MS/MS spectra, we use the 'MLP_BIN' encoder:

- The input spectra are represented as bins of mass-to-charge (m/z) ratios and intensities.

- The 'MLP_BIN' model processes these binned inputs through multiple fully connected layers, where each layer applies a ReLU activation and dropout to prevent overfitting.

- The output of the 'MLP_BIN' encoder is a vector representing the spectral data in an embedding space suitable for contrastive learning.

**Interaction model.**    Once the molecular and spectral embeddings are computed, they are concatenated and passed through an interaction MLP, which predicts the interaction score between the molecule and the scaffold. The interaction score is used to rank candidate scaffolds. The molecular encoder and spectral encoder are trained jointly in a contrastive learning framework, where the goal is to align the embeddings of correct molecular-scaffold pairs.

### A.1.2    Stage 2: Scaffold-Conditioned Molecular Generation

In Stage 2, the retrieved scaffold is used as the foundation for generating the full molecular structure, guided by the mass spectra data. This stage employs a **Graph Transformer** to integrate node, edge, and spectral features iteratively across multiple layers.

**Input representation.**    The inputs to the Graph Transformer in this stage consist of:

- **Node Features ($V$):** Each node represents an atom in the molecular scaffold, and the feature vector encodes atom type and properties.

- **Edge Features ($E$):** Bonds between atoms in the scaffold are represented as edge features.

- **Spectral Features ($S$):** The MS/MS spectra provide pairs of mass-to-charge (m/z) ratios and intensities.

**Multi-head attention.** Each layer of the Graph Transformer applies a **Node-Edge Block**, where both node and edge features are updated using attention mechanisms.

- **Self-Attention:** The model computes queries, keys, and values for each node and edge, allowing it to focus on relevant parts of the molecular graph during the update process.

- **Cross-Attention:** Cross-attention between the node/edge features and the spectral features enables the generation process to be conditioned on the spectral data, ensuring that the generated molecular structure aligns with the spectra.

**Feedforward networks.** After the attention layers, a **FeedForward Network** processes the updated node and edge features, further refining the representations.

**Layer normalization and residual connections.** Each attention block is followed by **Layer Normalization** and residual connections to stabilize training and maintain information flow across the layers.

**Final output.** After the final transformer layer, the updated node and edge features are passed through an output MLP to generate the final molecular structure. This process ensures that the generated molecule is consistent with both the scaffold and the spectral data.

## A.2 TRAINING HYPERPARAMETERS

The model is trained with a batch size of 64 and employed 47 workers for data loading. The learning rate is set to $2 \times 10^{-4}$, while weight decay is configured at $1 \times 10^{-12}$. Training proceeds for 2000 epochs, with the model logging progress every 40 steps.

A Markov bridge process with 100 steps is employed during training, and a cosine noise schedule is employed.

The model consists of 5 layers, with node, edge, and spectral features set at 64 dimensions each. The MLP hidden dimensions are configured to 256 for nodes, 128 for edges, and 256 for spectral features. The model also employs 8 attention heads for cross-attention and self-attention mechanisms. The feedforward dimensions are set to 256 for nodes, 128 for edges, and 128 for global features. This architecture enables efficient handling of both molecular structure and spectral data during training.

## A.3 VARIATION DISTRIBUTION AND ELBO

We first show the full derivation of the ELBO, which introduces a forward transition distribution $p(e_{t+1}|e_t, e_T)$ as the variational distribution. Then we discuss the formulation of the variational

distribution. The derivation of the ELBO is as follow:

$$\log p_\theta(G|S) = \log \sum_{e_1:e_{T-1}} \prod_{t=0}^{T-1} p_\theta(e_{t+1}|e_t, \mathcal{E}^S, \mathcal{V}^G) \tag{9}$$

$$= \log \sum_{e_1:e_{T-1}} \frac{p(e_{1:T-1}|e_0, e_T)}{p(e_{1:T-1}|e_0, e_T)} \prod_{t=0}^{T-1} p_\theta(e_{t+1}|e_t, \mathcal{E}^S, \mathcal{V}^G) \tag{10}$$

$$\geq \mathbb{E}_{p(e_1:e_{T-1}|e_0, e_T)}\left[\log \frac{\prod_{t=0}^{T} p_\theta(e_{t+1}|e_t, \mathcal{E}^S, \mathcal{V}^G)}{p(e_{0:T-1}|e_T)}\right] \tag{11}$$

$$= \mathbb{E}_{p(e_0:e_{T-1}|e_0, e_T)}\left[\sum_{t=0}^{T} \log \frac{p_\theta(e_{t+1}|e_t, \mathcal{E}^S, \mathcal{V}^G)}{p(e_{t+1}|e_t, e_T)}\right] \tag{12}$$

$$= \sum_{t=0}^{T-1} \mathbb{E}_{p(e_t, e_{t+1}|e_0, e_T)}\left[\log \frac{p_\theta(e_{t+1}|e_t, \mathcal{E}^S, \mathcal{V}^G)}{p(e_{t+1}|e_t, e_T)}\right] \tag{13}$$

$$= \sum_{t=0}^{T-1} \mathbb{E}_{p(e_t|e_0, e_T)}\left[\mathbb{E}_{p(e_{t+1}|e_t, e_T)} \log \frac{p_\theta(e_{t+1}|e_t, \mathcal{E}^S, \mathcal{V}^G)}{p(e_{t+1}|e_t, e_T)}\right] \tag{14}$$

$$= \sum_{t=0}^{T-1} \mathbb{E}_{p(e_t|e_0, e_T)}\left[-\text{KL}\Big(p(e_{t+1}|e_t, e_T)\big\|p_\theta(e_{t+1}|e_t, \mathcal{E}^S, \mathcal{V}^G)\Big)\right] \tag{15}$$

$$= -T\mathbb{E}_{\mathcal{U}(t;0,T-1)}\mathbb{E}_{p(e_t|e_0, e_T)}\left[\text{KL}\Big(p(e_{t+1}|e_t, e_T)\big\|p_\theta(e_{t+1}|e_t, \mathcal{E}^S, \mathcal{V}^G)\Big)\right] \tag{16}$$

$$:= \mathcal{L}_\theta(S, G) \tag{17}$$

The forward distribution defines a distribution of trajectories $e_{1:T-1}$ between $e_0$ and $e_T$. Note that $e_0$ is always 0 (non-edge) and is independent from $e_T$, so we have

$$p(e_{0:T-1}|e_T) = p(e_{1:T-1}|e_0, e_T). \tag{18}$$

This also satisfies the Markov property

$$p(e_{0:T-1}|e_T) = \prod_{t=0}^{T-1} p(e_{t+1}|e_t, e_T) = \prod_{t=0}^{T-1} \text{Categorical}(e_{t+1}; \mathbf{Q}_t(e_T)e_t). \tag{19}$$

The transition matrices $\mathbf{Q}_0, \ldots, \mathbf{Q}_{T-1}$ are $D \times D$ matrices, where

$$\mathbf{Q}_t(e_T) = \alpha_t \mathbf{I}_D + (1 - \alpha_t)e_T \mathbf{I}_D^T. \tag{20}$$

$\alpha_0, \ldots, \alpha_{T-1}$ are scheduling parameters similar to Austin et al. (2021). And $\mathbf{I}_D$ is a $D \times D$ identity matrix. With the defined transition matrices, the one-step transition probability $p(e_t|e_0, e_T)$ also has the closed form

$$p(e_t|e_0, e_T) = \text{Categorical}(e_{t+1}; \bar{\mathbf{Q}}_t(e_T)e_t), \quad \bar{\mathbf{Q}}_t(e_T) = \prod_{\tau=0}^{t} \mathbf{Q}_\tau(e_T) \tag{21}$$

Now that both $p(e_t|e_0, e_T)$ and $p(e_{t+1}|e_t, e_T)$ can be derived in closed-form, we can directly optimize the ELBO $\mathcal{L}_\theta(S, G)$.

## A.4 OVERALL WORKFLOW

The two stages work together to form a scaffold-conditioned molecular generation system. In the first stage, the model retrieves a scaffold using contrastive learning and the 'MLP_BIN' spectral encoder, and in the second stage, the Graph Transformer uses this scaffold to generate a complete molecule, conditioned on the spectral data. This two-step approach ensures that the molecular generation process is both accurate and guided by experimentally observed spectra.

