# OpenReview forum: "MADGEN: Mass-Spec attends to De Novo Molecular generation"
_ICLR.cc/2025/Conference — ICLR 2025 Poster_

### Official Review · Reviewer_HW4z · 2024-11-02

**Soundness:** 3
**Presentation:** 3
**Contribution:** 3
**Rating:** 6
**Confidence:** 3

**Summary:**

The paper presents MADGEN, a method for de novo molecular structure generation using MS/MS data to address challenges in spectral annotation. MADGEN operates in two stages: scaffold retrieval via contrastive learning and attention-based generation using the MS/MS spectrum. Evaluated on NIST23, CANOPUS, and MassSpecGym datasets, MADGEN shows strong performance, particularly with an oracle retriever, improving annotation by leveraging spectral data effectively.

**Strengths:**

1. The task of de novo generation of molecular structure from mass spectrum is important and challenging.  However, the AI community has not paid enough attention to this task.
2. The two-stage molecule structure generation method is novel.

**Weaknesses:**

1. The authors have not compared with previous de novo molecular elucidation methods from MS such as MSNovelist [1], Spec2Mol [2], MIST [3], etc.
2. manuscript’s presentation requires significant improvement for publication readiness:
    (1) Line 14 - Line 50: Double quotation marks should be directional.
    (2) Line 242 - Line 243: Why the word 'Following' is underlined?
    (3) Why there is $T$ in Eq. (6)?
    (4) \citep is used for parenthetical citations while \citet is used for textual citations, where the citation is part of the sentence. The authors are suggested to use these two commands appropriately to ensure clarity in their references and maintain consistency in citation formatting.
    (5) Line 102: The “F” in “Generative Frameworks for molecular generation” should be lowercase.
3. The code is not provided for reproducing the results.

[1]. MSNovelist: de novo structure generation from mass spectra (Nature Methods)
[2]. An end-to-end deep learning framework for translating mass spectra to de-novo molecules (Communications Chemistry)
[3]  Annotating metabolite mass spectra with domain-inspired chemical formula transformers (Nature Machine Intelligence)

**Questions:**

See the weakness above.

**Details Of Ethics Concerns:**

There is no ethics concern.

---

> ### Author Response · Authors · 2024-11-23
> **Response**
>
> **Question 1**: The authors have not compared with previous de novo molecular elucidation methods from MS such as MSNovelist [1], Spec2Mol [2], MIST [3], etc.
>
> **Response**: Thank you for the suggestion! We have included MSNovelist and Spec2Mol in our experiments. We report Top-1/Top-10 accuracy, similarity, and MCES in Table 2 in the updated draft. Note that for MSNovelist, only the accuracy is available.
>
> We did not include MIST in our comparisons, as it predicts molecular fingerprints rather than performing de novo molecular structure generation, which is not the focus of our method, of MSNovelist, and of Spec2Mol.
>
> ---
>
> **Question 2**: manuscript’s presentation requires significant improvement for publication readiness
>
> **Response**:  Thank you for pointing out these issues.  We have fixed them in the paper and optimized the presentation of the paper.
> - Line 242 - Line 243: Why the word 'Following' is underlined
>
> This is a rendering problem in the OpenReview when using \citet. We found that Following is underlined when using google scholar pdf viewer but not google chrome viewer.
>
> - Why there is T in Eq. (6)
>
> The T means the endpoint of the bridge process - we have defined $\mathcal{E}_T:=\mathcal{E}$ in line 239, and $e_T$ is an element in $\mathcal{E}_T$/
>
> - \citep is used for parenthetical citations while \citet is used for textual citations, where the citation is part of the sentence. The authors are suggested to use these two commands appropriately to ensure clarity in their references and maintain consistency in citation formatting.
>
> We have double-checked the \citep and \citet in our draft and fixed all incorrect usage the issues.
>
> - Line 102: The “F” in “Generative Frameworks for molecular generation” should be lowercase.
>
> Fixed
>
> ---
> **Question 3**: The code is not provided for reproducing the results.
>
> **Response**: Thanks for the comment. We'd provided the anonymous repository in here: https://anonymous.4open.science/r/abc-482F. Note that our code is built upon the repository https://github.com/igashov/RetroBridge.
>
> ---
>
> [1]. MSNovelist: de novo structure generation from mass spectra (Nature Methods)
>
> [2]. An end-to-end deep learning framework for translating mass spectra to de-novo molecules (Communications Chemistry)
>
> [3]. Annotating metabolite mass spectra with domain-inspired chemical formula transformers (Nature Machine Intelligence)

---

> ### Comment · Reviewer_HW4z · 2024-11-30
> **Thanks for the response and additional experiments!**
>
> Thank you for the response and additional experiments. My concerns have been addressed, and I have adjusted my score to 6 to support this work. I believe that de novo molecule generation from mass spectra is a highly important problem in practice, yet it has not received sufficient attention from the AI research community. The authors’ idea of using retrieved molecular scaffolds to guide molecule structure elucidation is also interesting.

---

> > ### Author Response · Authors · 2024-11-30
> > **Thanks for the feedback!**
> >
> > Thank you for your thoughtful feedback and for taking the time to review our work. We greatly appreciate your comments on the significance of de novo molecule generation from mass spectra and the novelty of our approach. We are glad that our additional experiments addressed your concerns. Your support and constructive input have been invaluable in refining our work. Thank you again for your time and effort in reviewing our paper.

---

### Official Review · Reviewer_RWgk · 2024-11-02

**Soundness:** 3
**Presentation:** 4
**Contribution:** 3
**Rating:** 6
**Confidence:** 3

**Summary:**

The paper introduces MADGEN, a framework for de novo molecular structure generation from mass spectrometry data. MADGEN employs a two-stage approach: first, it retrieves a molecular scaffold, and second, it completes the molecule conditioned on both the scaffold and the MS/MS spectra. Evaluated on datasets like NIST23, CANOPUS, and MassSpecGym, MADGEN effectively reduces search complexity and enhances accuracy.

**Strengths:**

1. The two-stage approach of scaffold retrieval followed by scaffold-conditioned molecular generation presents a novel solution for de novo molecular structure prediction.
2. The paper is well-written and easy to follow.
3. The model is evaluated on multiple datasets, and a detailed ablation study is provided.

**Weaknesses:**

1. The scaffold retrieval performance, especially when using a predictive retriever, remains relatively low (e.g., NIST23).
2. The discussion of baselines is unclear.

**Questions:**

1. In Section 3.2.1, the authors state, “Since the atom set V can be directly inferred from the chemical formula.” However, it is unclear where the chemical formula is derived from. Could the authors clarify this, as in real-world scenarios, the input typically does not contain the chemical formula?
2. For the MassSpecGym dataset, what is the “best published state-of-the-art performance” referred to? Are there baseline performance metrics reported for the other two datasets as well?

---

> ### Author Response · Authors · 2024-11-23
>
> **Question 1**: In Section 3.2.1, the authors state, “Since the atom set V can be directly inferred from the chemical formula.” However, it is unclear where the chemical formula is derived from. Could the authors clarify this, as in real-world scenarios, the input typically does not contain the chemical formula?
>
> **Response**: Chemical formulas are derived based on the MS1 peak, which specifies the molecular weight of the ionized molecule that becomes fragmented and measured as a spectrum. From the  weight of the ionized molecule, one could generate candidate formulas.  Additional information such as the spectrum can be used to refine the list of candidate formulas.
>
> For example, in the recent MIST-CF[1], a dynamic programming algorithm generates exhaustive chemical formula candidates for the MS1 peak within a small mass tolerance, often filtering implausible options using chemical rules like ring double bond equivalents (RDBE).  Subsequently, peaks within the spectrum are annotated with subformulas using a Formula Transformer, a neural network, based on the candidate formulas. MIST-CF scores each candidate formula based on its alignment with the observed fragmentation spectrum, outputting a ranked list of likely formulas.  SIRIUS[2] is another de noo tool that assigns a chemical formula to a spectrum.  Like, MIST-CF, SIRIUS generates candidate formulas, and assigns potential subformulas to peaks. These subformula annotations are then organized into a fragmentation tree using maximum a posteriori (MAP) optimization.  Finally, SIRIUS calculates the likelihood of each chemical formula based on the constructed tree. Another technique, BUDDY[3], each molecular weight associated with each  peak within the spectrum is searched against a curated molecular formula database. Similarly, the neutral loss (what was lost during fragmentation and not measured) is also searched in the formula database. BUDDY prioritizes explainable candidate formulas and filters implausible formulas.  Currently, SIRIUS and BUDDY are the two most common tools used by practitioners to annotate their spectra.
>
> ---
>
> **Question 2**: For the MassSpecGym dataset, what is the “best published state-of-the-art performance” referred to? Are there baseline performance metrics reported for the other two datasets as well? **& Weakness 4**:  The discussion of baselines is unclear.
>
> **Response**: The manuscript describing the MassSpecGym dataset was just accepted at NeurIPs 2024 [4]. For de novo generation with a known chemical formula, performance is reported for “random chemical generation”, “SMILES transformer”, and a “SELFIES Transformer”.  The accuracy was zero for all three techniques.  We also report on running additional tools, namely Spec2Mol[5] and MSNovelist[6], as summarized in Table 2.
>
> Performance metrics include Top-1 accuracy, similarity, and MCES (Maximum Common Edge Subgraph). The results indicate that Top-1 accuracy was zero across all datasets, but similarity and MCES scores varied. For example, Spec2Mol achieved higher similarity and MCES values on MassSpecGym compared to NIST23 and Canopus, with similarity scores of 0.19 (MassSpecGym), 0.16 (NIST23), and 0.18 (Canopus). Corresponding MCES values were 45.89 (MassSpecGym), 20.88 (NIST23), and 38.97 (Canopus).
>
> ---
>
> **Weakness 3**: The scaffold retrieval performance, especially when using a predictive retriever, remains relatively low (e.g., NIST23).
>
> **Response**: We have double-checked the implementation for NIST23 and noticed there is a numerical issue when computing the cosine similarity score. We fix the bug and re-do the experiment for NIST23. Correctness of other datasets are also double-checked, below is the updated result for NIST23:
> |Retriever|SPA↑|Top1 Accuracy↑|Top1 Similarity↑|Top1 MCES↓|Top10 Accuracy↑|Top10 Similarity↑|Top10 MCES↓ |
> |-|-|-|--|-|-|-|-|
> |MADGEN (Predictive) (Before)| 8.7%|1.8%|  0.06| 84.24 |2.2% |0.07|82.33|
> |MADGEN (Predictive) (Now)| **57.8%** |**10.3%**| **0.18**| **68.13** |**14.5%**|**0.24**|**62.65**|
>
> We hope the updated result address your concern.
>
> ---
>
> **Reference**
> #### [1] Goldman, Samuel, et al. "MIST-CF: Chemical formula inference from tandem mass spectra." Journal of Chemical Information and Modeling 64.7 (2023): 2421-2431.
> #### [2] Dührkop, Kai, et al. "SIRIUS 4: a rapid tool for turning tandem mass spectra into metabolite structure information." Nature methods 16.4 (2019): 299-302.
> #### [3] Xing, Shipei, et al. "BUDDY: molecular formula discovery via bottom-up MS/MS interrogation." Nature Methods 20.6 (2023): 881-890.
> #### [4] Bushuiev, Roman, et al. "MassSpecGym: A benchmark for the discovery and identification of molecules." arXiv preprint arXiv:2410.23326 (2024).
> #### [5] Litsa, Eleni, et al. "Spec2Mol: An end-to-end deep learning framework for translating MS/MS Spectra to de-novo molecules." (2021).
> #### [6] Stravs, Michael A., et al. "MSNovelist: de novo structure generation from mass spectra." Nature Methods 19.7 (2022): 865-870.

---

> > ### Author Response · Authors · 2024-12-03
> > **Final response**
> >
> > Dear reviewer,
> >
> > I hope we have addressed your concerns including the baseline and the scaffold retrieval performance on NIST23. Your feedback is highly valued.
> >
> > Thank you for your time.
> >
> > Best regards,
> >
> > The authors

---

### Official Review · Reviewer_t2EJ · 2024-11-03

**Soundness:** 3
**Presentation:** 3
**Contribution:** 3
**Rating:** 6
**Confidence:** 2

**Summary:**

This paper introduces MADGEN, a method for de novo molecular generation using mass spectrometry data. MADGEN simplifies the structure generation process by retrieving molecular scaffolds and building complete molecules upon them. The experimental results on three datasets demonstrate that MADGEN can effectively generate accurate molecular structures when the scaffold is known.

**Strengths:**

- The use of scaffolds for simplifying molecular generation is a novel and effective strategy that reduces complexity.
- The paper is clear and easy to understand.

**Weaknesses:**

- The method lacks a comparison with other baselines.

**Questions:**

- I would appreciate it if you could include a comparison of MADGEN with other prediction methods.

---

> ### Author Response · Authors · 2024-11-23
> **Response to lack of baselines**
>
> Thanks for the suggestion, we have added baselines (Spec2Mol [1], MSNovelist [2], Random Chemical Generator [3], SMILES Transformer [3], SELFIES Transformer [3])  in the paper. For your convenience, we have attached the table below, which can also be found in the updated draft.
> | Retriever              | SPA↑    | Top1 Accuracy↑ | Top1 Similarity↑ | Top1 MCES↓ | Top10 Accuracy↑ | Top10 Similarity↑ | Top10 MCES↓ |
> |------------------------|---------|----------------|------------------|------------|-----------------|-------------------|-------------|
> | **NIST**              |         |                |                  |            |                 |                   |             |
> | Spec2Mol              | -       | 0.0%           | 0.16             | *20.88*    | 0.0%           | 0.20             | 13.66       |
> | MSNovelist            | -       | 0.0%           | -                | -          | 0.0%           | -                | -           |
> | MADGEN\_Pred.         | 57.8%   | *10.3%*        | *0.18*           | 68.13      |    	14.5%|	0.24|	62.65|
> | MADGEN\_Oracle        | 100%    | **49.0%**      | **0.69**         | **18.48**  | **65.5%**      | **0.85**         | **3.88**    |
> | **CANOPUS**           |         |                |                  |            |                 |                   |             |
> | Spec2Mol              | -       | 0.0%           | *0.18*           | **38.97**  | 0.0%           | 0.26             | **23.97**   |
> | MSNovelist            | -       | 0.0%           | -                | -          | 0.0%           | -                | -           |
> | MADGEN\_Pred.         | 37.9%   | *1.0%*         | 0.14             | 70.45      | *1.0%*         | *0.51*           | 45.61       |
> | MADGEN\_Oracle        | 100%    | **8.9%**       | **0.25**         | *67.91*    | **14.7%**      | **0.87**         | *36.36*     |
> | **MSGym**             |         |                |                  |            |                 |                   |             |
> | Rand. Gen.            | -       | 0.0%           | 0.08             | **21.11**  | 0.0%           | 0.11             | *18.25*     |
> | SMILES Transformer    | -       | 0.0%           | 0.03             | 79.39      | 0.0%           | 0.10             | 52.13       |
> | SELFIES Transformer   | -       | 0.0%           | 0.08             | 38.88      | 0.0%           | 0.13             | 26.87       |
> | Spec2Mol              | -       | 0.0%           | *0.19*           | 45.89      | 0.0%           | *0.28*           | 32.60       |
> | MSNovelist            | -       | 0.0%           | -                | -          | 0.0%           | -                | -           |
> | MADGEN\_Pred.         | 34.8%   | *0.8%*         | 0.13             | 74.19      | *1.6%*         | 0.25             | 53.50       |
> | MADGEN\_Oracle        | 100%    | **18.8%**      | **0.61**         | *27.79*    | **38.6%**      | **0.87**         | **3.97**    |
>
> We hope this will address your concern.
>
> **References**
> #### [1] Litsa, Eleni, et al. "Spec2Mol: An end-to-end deep learning framework for translating MS/MS Spectra to de-novo molecules." (2021).
> #### [2] Stravs, Michael A., et al. "MSNovelist: de novo structure generation from mass spectra." Nature Methods 19.7 (2022): 865-870.
> #### [3] Bushuiev, Roman, et al. "MassSpecGym: A benchmark for the discovery and identification of molecules." arXiv preprint arXiv:2410.23326 (2024).

---

### Official Review · Reviewer_H3Zi · 2024-11-04

**Soundness:** 3
**Presentation:** 3
**Contribution:** 3
**Rating:** 6
**Confidence:** 4

**Summary:**

This study presents MADGEN, a two-stage framework for generating molecular structures from MS/MS data. In the first stage, MADGEN retrieves a scaffold using either predictive retrieval or oracle retrieval. In predictive retrieval, MADGEN treats scaffold selection as a ranking task, using contrastive learning to align embeddings of mass spectra and scaffold candidates in a shared latent space, scoring each scaffold to identify the best match. Oracle retrieval, by contrast, directly uses RDKit to extract the correct scaffold from a molecular graph. In the second stage, starting from the retrieved scaffold, MADGEN generates the full molecular structure through a Markov bridge-based expansion, sequentially adding atoms and bonds with classifier-free guidance to integrate spectral information. Evaluations were performed on datasets NIST23, CANOPUS, and MassSpecGym, underscoring its potential in metabolomics and drug discovery applications.

**Strengths:**

* The two-stage idea is interesting.
* The oracle retrieval method is more effective.

**Weaknesses:**

* The SPA of the predictive retrieval is very low.
* The predictive retrieval approach yields poor molecule generation in Phase 2, where the generated structures fail to align with target properties, underscoring a critical limitation.
* The conditioning of molecular generation on mass spectrometry data is largely based on classifier-free guidance, a well-established technique. The novelty is not well articulated.

**Questions:**

* How is the candidate scaffold pool determined for predictive retrieval? There can be a huge number of scaffold candidates.
* How does the contrast learning work for aligning the embeddings of mass spectra with their corresponding molecule? Some explanations are needed.
* Regarding molecular generation in Phase 2, the absorbing transition matrix only applies to isolated atoms, strictly enforcing scaffold structure. Will such a hard constraint be problematic especially when the chosen scaffold is incorrect?
* Oracle retrieval requires both the MS/MS spectrum and the chemical formula (molecular graphs). If chemical formula is known, what are the remaining challenging in solving the molecular structures?

---

> ### Author Response · Authors · 2024-11-23
> **Response**
>
> We thank the reviewer for the insightful questions, below we addressed the raised concerns and comments.
>
> ---
>
> **Question 1**: How is the candidate scaffold pool determined for predictive retrieval? There can be a huge number of scaffold candidates.
>
> **Response**: For NIST23 and CANOPUS, we selected all candidates from PubChem by providing the chemical formula. MassSpecGym provided 256 candidates per test molecule. We are not aiming to find the correct candidate, but the correct scaffold, which could reduce the number of candidates to retrieval. We will update the draft to provide more background knowledge about the candidate pool selection.
>
> ---
>
> **Question 2**: How does the contrast learning work for aligning the embeddings of mass spectra with their corresponding molecule? Some explanations are needed.
>
> **Response**: We consider a contrastive learning framework similar to CLIP [1], which aligns the embeddings from two modalities. Here we treat spectrum as one modality and scaffold as the other. The CLIP-based framework has been popularized for information retrieval framework [2-7], where one can use the embedding similarity to determine what’s the most likely paired item (scaffold) based on the query (spectrum). We will clarify the background and previous paradigm in our updated draft.
>
> ---
> **Question 3**: Regarding molecular generation in Phase 2, the absorbing transition matrix only applies to isolated atoms, strictly enforcing scaffold structure. Will such a hard constraint be problematic especially when the chosen scaffold is incorrect?
>
> **Response**: Thank you for the insightful question! Indeed, a typical absorbing diffusion transition does not support modifying what's generated. However, compared to uniform transition matrix, absorbing diffusion significantly reduces the modeling complexity. It is commonly shown by previous research [8] that absorbing diffusion outperforms uniform diffusion. Moreover, the mentioned "hard constraint" is amendable by further introducing solver such as predictor-corrector during sampling [9-11]. As we do not innovate the fundamental framework of diffusion models, we do not include such augmentation in our experiment or method.
>
> ---
>
> **Question 4**: Oracle retrieval requires both the MS/MS spectrum and the chemical formula (molecular graphs). If a chemical formula is known, what are the remaining challenges in solving the molecular structures?
>
> **Response**: A chemical formula (not molecular graph) can have multiple corresponding molecular structures due to many possible molecular arrangements. For example, there are 44,374 known molecular structures (and hence graphs) in the PubChem database associated with C12H18N2O2. There are also likely many more structures undocumented in any databases. So the challenge is realizing the exact molecular structure that gave rise to the MS/MS spectrum.
>
> ---
>
> **Weakness 5**: The SPA of the predictive retrieval is very low.
>
> **Response**: After submitting the draft, we notices that there is an unexpected issue in the implementation of stage 1 for NIST23 dataset. We fix the implementation to have a better SPA result for this dataset. We also check the correctness of other implementations. Specifically, the new result yields a SPA of 57.8% for NIST23, 37.9% for CANOPUS, 34.8% for MSGym. We kindly refer the reviewer to the revised draft for the more details.
>
>  ---
>
> **Weakness 6**: The predictive retrieval approach yields poor molecule generation in Phase 2, where the generated structures fail to align with target properties, underscoring a critical limitation.
>
> **Response**: We have introduced various baseline result in our updated draft. Specifically, our retrieval accuracy outperforms all baselines that are reproducible. We'd like again emphasize the challenging of MS/MS spectrum annotation task. To the best of our knowledge, our method is currently the SOTA in this venue.
>
> ---
>
> **Weakness 7**: The conditioning of molecular generation on mass spectrometry data is largely based on classifier-free guidance, a well-established technique. The novelty is not well articulated.
>
> **Response**:  Thanks for the comment. While we employ CFG in our methodology for guided generation, we do not claim such a technique as our major contribution. Specifically, our major contribution is to propose a two-stage generative retrieval framework for mass spectra annotation(area). Under the proposed framework, we explore various implementations of CFG that give better performance.The implementation, which involves the spectrum embedding module and spectrum-molecule-interaction module, are novel to the community.

---

> > ### Author Response · Authors · 2024-11-23
> > **Cont. Response**
> >
> > **References:**
> > #### [1] Radford, Alec, et al. "Learning transferable visual models from natural language supervision." International conference on machine learning. PMLR, 2021.
> > #### [2] Luo, Huaishao, et al. "Clip4clip: An empirical study of clip for end to end video clip retrieval." arXiv preprint arXiv:2104.08860 (2021).
> > #### [3] Lei, Jie, et al. "Less is more: Clipbert for video-and-language learning via sparse sampling." Proceedings of the IEEE/CVF conference on computer vision and pattern recognition. 2021.
> > #### [4] Bain, Max, et al. "A clip-hitchhiker's guide to long video retrieval." arXiv preprint arXiv:2205.08508 (2022).
> > #### [5] Fang, Han, et al. "Clip2video: Mastering video-text retrieval via image clip." arXiv preprint arXiv:2106.11097 (2021).
> > #### [6] Ma, Yiwei, et al. "X-clip: End-to-end multi-grained contrastive learning for video-text retrieval." Proceedings of the 30th ACM International Conference on Multimedia. 2022.
> > #### [7] Hendriksen, Mariya, et al. "Extending CLIP for Category-to-image Retrieval in E-commerce." European Conference on Information Retrieval. Cham: Springer International Publishing, 2022.
> > #### [8] Austin, Jacob, et al. "Structured denoising diffusion models in discrete state-spaces." Advances in Neural Information Processing Systems 34 (2021): 17981-17993.
> > #### [9] Campbell, Andrew, et al. "A continuous time framework for discrete denoising models." Advances in Neural Information Processing Systems 35 (2022): 28266-28279.
> > #### [10] Campbell, Andrew, et al. "Generative flows on discrete state-spaces: Enabling multimodal flows with applications to protein co-design." arXiv preprint arXiv:2402.04997 (2024).
> > #### [11] Lezama, Jose, et al. "Discrete predictor-corrector diffusion models for image synthesis." The Eleventh International Conference on Learning Representations. 2022.

---

> > ### Comment · Reviewer_H3Zi · 2024-12-01
> > **Thanks authors for their efforts.**
> >
> > I increased my score to 6.

---

> ### Author Response · Authors · 2024-12-01
> **Thanks for your feedback!**
>
> We sincerely appreciate you taking the time to reconsider our work and increasing your score. Your effort and support in evaluating our submission mean a great deal to us.
>
> Best,
>
> The Authors

---

### Author Response · Authors · 2024-11-23
**General response**

We thank all reviewers for your time in reviewing our submission and your valuable comments that help improve our work. We have made changes accordingly, which can be found in the updated draft as well as the individual response to each reviewer. Here we’d like to summarize the changes based on the suggestions. We further clarify the addressed task and motivation, and summarize the contribution(novelty and performance) of MADGEN.

## Changes

---
- **Including more baselines**: We added comparisons with additional state-of-the-art methods, including MSNovelist[1] and Spec2Mol[2], and provided detailed performance metrics (e.g., Top-1 accuracy, similarity, and MCES) across datasets. These comparisons highlight the consistent improvement of our method over existing techniques. **(per t2EJ, RWgk, HW4z)**
- **Improving Predictive retrieval performance on NIST23**: We identified and corrected a numerical issue in the NIST23 scaffold retrieval experiments, leading to improved results that are now included in the manuscript. **(per H3Zi, RWgk)**
- **Paper presentation**: We enhanced the clarity and readability of the manuscript by addressing all editorial suggestions, fixing typographical errors, and expanding explanations of our methodology and results. **(per HW4z)**
- **Code upload**: We provide the code implementation via https://anonymous.4open.science/r/abc-482F **(per HW4z)**

---

## Contributions

- **Task and motivation**: Mass spectra annotation is a critical task in fields such as metabolomics, drug discovery, and environmental analysis. Accurately annotating mass spectra enables researchers to identify molecular structures from spectral data, facilitating the discovery of novel compounds and the characterization of biochemical pathways. Despite its importance, the task remains highly challenging due to the vast chemical space, the ambiguity of molecular fragmentations, and the lack of annotated spectral databases for many compound classes. Scaffold retrieval plays a vital role in this process by narrowing down the candidate space and guiding downstream molecular generation tasks. By identifying the correct scaffold, our approach lays a strong foundation for generating plausible molecular structures, thereby advancing the broader goal of automating mass spectra annotation.

- **Novelty and performance**: MADGEN aims at the challenging task of mass spectra annotation, focusing specifically on scaffold retrieval as a key component. The primary innovation of our approach is the introduction of a two-stage generative retrieval framework, which separates scaffold retrieval (Stage 1) and scaffold-based molecular generation (Stage 2). This design allows us to explore architectural innovations for embedding alignment and spectrum-molecule interaction. While the overall accuracy for the task is low due to the inherent difficulty of mass spectra annotation, our method achieves significantly better performance compared to existing baselines, showcasing its effectiveness and potential in tackling this complex problem.

---

### Author Response · Authors · 2024-11-30
**Thanks for your time and feedbacks!**

Dear Reviewers,

Thank you once again for taking the time to review our work and providing detailed comments and feedback.

During the rebuttal process, we have carefully addressed each of your points and hope that our responses have resolved your concerns. If there is anything that remains unclear or if you have further questions, we would be more than happy to discuss them.

As the discussion period is nearing its end, we kindly ask that you review our responses and let us know if they adequately address your concerns.

We sincerely appreciate your time and effort. Wishing you a wonderful day!

Best,
The Authors

---

### Meta-Review · Area_Chair_AjT6 · 2024-12-18

**Metareview:**

The paper introduces MADGEN, a two-stage framework for de novo molecular structure generation from MS/MS data, leveraging scaffold retrieval and attention-based molecular generation. It shows promise in metabolomics and drug discovery.

Strengths include the novel two-stage approach, effective use of scaffolds to simplify molecular generation, and strong performance with an oracle retriever.

The manuscript also requires improvement for publication readiness, and more clear discussion of baselines.

The decision to accept is based on the paper's novel approach to a significant challenge, strong performance in certain conditions, and potential impact on the field.

**Additional Comments On Reviewer Discussion:**

Reviewer H3Zi and Reviewer RWgk raised the problem that the scaffold retrieval performance remains relatively low. The authors corrected their implementation and updated their results.

Reviewer t2EJ, Reviewer RWgk, and Reviewer HW4z asked for more baselines. The authors provided more comparison with more baselines.

---

### Decision · Program_Chairs · 2025-01-22

Accept (Poster)